# Molecular Monitoring of EHV-1 in Silently Infected Performance Horses through Nasal and Environmental Sample Testing

**DOI:** 10.3390/pathogens11070720

**Published:** 2022-06-24

**Authors:** Nicola Pusterla, Samantha Barnum, Amy Young, Eric Mendonsa, Steve Lee, Steve Hankin, Skyler Brittner, Carrie J. Finno

**Affiliations:** 1Department of Medicine and Epidemiology, School of Veterinary Medicine, University of California, Davis, CA 95616, USA; smmapes@ucdavis.edu; 2Center for Equine Health, School of Veterinary Medicine, University of California, Davis, CA 95616, USA; ayoung@ucdavis.edu (A.Y.); cjfinno@ucdavis.edu (C.J.F.); 3Fluxergy, Irvine, CA 92618, USA; emendonsa@fluxergy.com (E.M.); slee@fluxergy.com (S.L.); 4Desert International Horse Park, Thermal, CA 92274, USA; steve@deserthorsepark.com (S.H.); skyler@deserthorsepark.com (S.B.); 5Department of Population Health and Reproduction, School of Veterinary Medicine, University of California, Davis, CA 95616, USA

**Keywords:** EHV-1, equine, silent shedder, nasal swab, environmental surveillance, qPCR

## Abstract

While the main goal in the management of an EHM outbreak focuses on identifying early clinical disease in order to physically separate infected horses, little effort is placed towards monitoring healthy horses. The assumption that EHV-1 shedding parallels clinical disease is erroneous, as subclinical shedders have been shown to be actively involved in viral spread. In an attempt to document the frequency of EHV-1 shedders and their impact on environmental contamination, we collected nasal swabs from 231 healthy horses and 203 environmental samples for the testing of EHV-1 by qPCR. Six horses and 28 stalls tested qPCR-positive for EHV-1. There was no association in the EHV-1 qPCR-positive status between nasal and stall swabs. While testing nasal secretions of healthy at-risk horses can detect active shedding at a specific time point, the testing of stall swabs allows to assess the temporal EHV-1 shedding status of a horse. The study results highlight the risk of subclinical EHV-1 shedders and stalls occupied by these horses as sources of infection for susceptible horses. The testing of individual stalls for the presence of EHV-1 may be a more practical approach than the collection of individual nasal swabs for the monitoring and early detection of the circulating virus. The results also highlight the need to improve the cleanliness and disinfection of stalls utilized by performance horses during show events.

## 1. Introduction

In recent years, large equestrian show venues have experienced EHV-1 outbreaks resulting in equine herpesvirus myeloencephalopathy (EHM) [1,2]. Various factors have been incriminated in the development of such EHM outbreaks, including the large number of competing horses and the increased movement of such horses during the shows. Specific factors, including the number of classes competed in at the event, female sex, biosecurity-related activities at events, and recent vaccination against EHV-1, have been associated with an increased risk for EHM [1,3]. Unfortunately, by the time horses are diagnosed with EHM, EHV-1 has spread often subclinically far beyond the index case [4]. The highly contagious nature of EHV-1 infected cases has been the demise of many show barns, racing venues, and veterinary hospitals [5,6,7,8,9,10]. Preventive strategies aiming to reduce the risk of an EHM outbreak have mainly focused on biosecurity protocols since, at this time, no vaccine is available with the claim to prevent EHM [11]. Unfortunately, biosecurity protocols often lack compliance, especially during non-critical times, which may predispose transmission of EHV-1 amongst horses. The rate of silent transmission (i.e., transmission without clinical disease) of EHV-1 during EHM outbreaks can reach as high as 27% [3,12,13]. This is in sharp contrast to the rate of shedding of EHV-1 amongst healthy equids outside an EHM outbreak, which ranges from 0–4%, depending on age and population tested [14,15,16,17,18,19]. Therefore, the aim of the present study was to determine the frequency of EHV-1 detection by qPCR in nasal secretions of healthy horses and their respective stalls. 

## 2. Results

Nasal swabs were collected from 231 adult, healthy sport horses 28 days after the beginning of an EHM outbreak at a large hunter/jumper show venue in southern California. The horses were stabled in 21 different barns, with a median number of 10 horses per barn (range 1–39 horses per barn). Because of confidentiality issues, age and sex of the 231 horses were not recorded. Six horses (2.6%) tested qPCR-positive for both the *gB* gene and the N_752_ genotype of EHV-1 at the genomic DNA level. The EHV-1 qPCR-positive horses originated from five different barns, and the viral load ranged from 62 to 670,939 EHV-1 *gB* genes/million cells (median 55,597 EHV-1 *gB* genes/million cells; Figure 1 and Table 1). Five of the EHV-1 qPCR-positive horses originated from four barns (barn 7, 32, 39, and 45) with no previously diagnosed EHV-1 cases, while one horse was stabled in a barn (barn 1) with previously diagnosed EHV-1 cases. 

Environmental swabs were collected from 203 stalls from 16 different barns. The samples originated from 189 stalls occupied by a horse and from 14 empty stalls with horses recently diagnosed with clinical EHV-1 infection and subsequently moved to isolation. Time from horse moved to isolation to environmental sample collection ranged from 1 to 15 days (median 8 days). Twenty-eight stalls (13.8%) tested qPCR-positive for both the *gB* gene and the N_752_ genotype of EHV-1 at the genomic DNA level. Twenty-six EHV-1 qPCR-positive environmental swabs were taken in stalls of horses with negative EHV-1 qPCR results in nasal secretions. Two additional EHV-1 qPCR-positive environmental swabs were taken from empty stalls, in which horses with clinical EHV-1 infection were housed prior to being moved to isolation. There were more EHV-1 qPCR-positive stall swabs from barns with previously diagnosed clinical EHV-1 cases (barn 1, 2, 4, 32, and FEI II) as compared to barns with no reported EHV-1 cases (barn 7, 8, 32, 37, 39, 45, and FEI III). The absolute values for EHV-1 qPCR-positive stall swabs ranged from 2.64 to 14,275 *gB* genes/µL of purified DNA (median 66.7, Figure 1 and Table 1). 

## 3. Discussion

Investigations of EHM outbreaks rarely identify the point source of the infection. It is often speculated that either a subclinically infected horse or a horse experiencing reactivation of latency with subsequent active shedding is at the origin of an outbreak [1,2]. Unfortunately, by the time the first EHM index case is identified during an outbreak, EHV-1 has had plenty of time to spread amongst susceptible horses. One of the greatest challenges in reducing the spread of EHV-1 during an outbreak is to determine the shedding status of every single horse with direct or indirect contact with the index case. While the morbidity rate during outbreaks of EHM for horses with neurological and respiratory disease can be as high as 84% [12], little is known about the number of subclinically infected horses. It was, therefore, the aim of this study to investigate the frequency of EHV-1 detection in healthy horses quarantined during a large EHM outbreak. Because of the often short duration of EHV-1 shedding in subclinically infected horses, the authors also sought to investigate the environmental burden of EHV-1 in the stalls of healthy horses. 

The frequency of EHV-1 detection in 231 healthy adult sport horses was 2.6%, which is in the range of 0–4% previously reported for healthy horses not associated with an EHV-1 outbreak [14,15,16,17,18,19]. The frequency of EHV-1 detection in the study population was remarkably lower than the ones investigated in healthy horses during active EHM outbreaks [3,12,13]. The lower frequency of EHV-1 shedding in the study horses likely relates to the time of sample collection, as the nasal swabs were collected 28 days following the diagnosis of the first index case. Further, the isolation protocols and strict biosecurity protocols at the show grounds likely contributed to the reduced detection frequency of EHV-1. Many field studies have reported on the rapid reduction of EHV-1 shedding in clinically infected animals, with the majority of affected horses shedding the virus for 5–7 days after acute onset of disease [9,12,20]. While EHV-1 viral load in nasal secretions of subclinically infected horses has been reported to be highly variable, a few studies have shown no statistical difference in viral load between subclinical shedders and horses with fever and neurological disease [4,13,21,22]. Viral load in the nasal secretions of study horses ranged from 61 to 670,939 EHV-1 *gB* genes/million cells, which is similar to previous studies with ranges from 66 to 980,000 EHV-1 *gB* genes/million cells [13,22]. Viable virus, characterized by the presence of transcriptional activity of the *gB* gene, has previously been associated with viral loads ≥1 × 10^4^ *gB* genes/million cells at the genomic DNA level [21]. In the present study, four out of six EHV-1 qPCR-positive nasal secretions would have contained viable virus, representing a source of transmission to susceptible horses. However, no cell culture was performed on these nasal swabs to determine lytic EHV-1. 

EHV-1 qPCR results for stall swabs accurately reflected past EHV-1 shedding of subclinically and clinically infected horses. Interestingly, there was no association in the EHV-1 qPCR-positive status between nasal and stall swabs. The origin of EHV-1 is speculative but likely originated from a focal source and slowly spread thereafter directly via horse-to-horse contact or indirectly via personal or contaminated equipment. Previous studies have shown that silently infected horses shed EHV-1 for a short period, which could explain the inability to detect EHV-1 in the horses occupying the positive stalls [4]. While testing nasal secretions of healthy at-risk horses can detect active shedding at a specific time point, the testing of stall swabs allows assessing the temporal EHV-1 shedding status of a horse. The results showed that, overall, there were more EHV-1 qPCR-positive stalls than horses. Further, the number of EHV-1 qPCR-positive stalls was greater in barns with previously diagnosed clinical cases. The results show the impact of clinical disease on subclinical infection and secondary environmental contamination. EHV-1 has been shown to remain stable and infectious under various environmental conditions such as water for up to three weeks [23,24]. Further, a recent study determined that irrespective of environment-material evaluated (leather, polyester-cotton fabric, pinewood shavings, wheat straw, and plastic), viable virus could still be recovered at 48 h following standard inoculation with EHV-1 [25]. While DNA has been shown to be resistant to degradation in the environment, it remains to be determined how long nucleic acids from EHV-1 can be detected in the environment, such as stalls. Independent of the duration of EHV-1 detection, the presence of environmental EHV-1 should trigger measures aimed at cleaning and disinfecting stalls once they have been vacated and prior to new show horses using the stalls. Because of the cumulative effect of EHV-1 shedding in infected horses, the swabbing of stalls for qPCR testing may be a more practical way to monitor horses compared to testing individual nasal swabs. This situation would mostly apply to multi-day horse events with large numbers of competing horses and would by no means replace routine biosecurity protocols. Future studies are needed to refine the swabbing protocol in order to focus mostly on areas with the highest contamination load. Further, to reduce costs of testing, the pooling of environmental samples for the testing of EHV-1 should also be evaluated, similar to what is done for *Salmonella* spp. environmental testing [26].

Study limitations related to the single sample collection time point. The collection of multiple samples from the same horses and the collection during the early stage of the outbreak would likely have yielded a higher detection frequency of EHV-1. Further, the environmental results may only be interpreted in the context of the collection material and sampling protocol used and cannot be extrapolated to other swab types or collection protocols. 

## 4. Materials and Methods

### 4.1. Outbreak and Study Population

On 10 February 2022, a 12-year-old Warmblood gelding attending an equestrian event in Riverside County, California, developed acute onset of neurological deficits and was diagnosed with EHM. While the affected horse was immediately isolated at the event premise, an additional 29 horses were diagnosed with clinical EHV-1 infection within the subsequent 24-day period. The premise was put under mandatory quarantine and released on 24 March (41 days from the first clinical EHV-1 case). On March 10th and 11th, seven of the investigators (N.P., A.Y., E.M., S.L., S.H., S.B. (Skyler Brittner), and C.J.F.) visited the premise in order to collect nasal and environmental swabs. At the time of the visit, the showgrounds housed approximately 430 horses. The majority of the horses were kept in their assigned barn and stall, grouped by trainer, and allowed daily scheduled outdoor exercise. Five horses with previous EHV-1 diagnosis were kept in an isolation tent. Stringent biosecurity protocols were in place for each horse barn and each horse group (footbath, personal protective equipment). All horses were monitored daily for respiratory and neurological signs, and rectal temperature was recorded twice daily. Only healthy horses and their environment were enrolled in the study. Study participation was voluntary, and informed written consent was available for each study horse.

### 4.2. Sample Collection and Analysis

Each study horse had a 6″ flocked swab (Avantik, Pine Brook, NJ, USA) collected either from the right or from the left rostral nasal passages. Each swab was placed in a vial containing 3 mL of the viral transport medium (Avantik, Pine Brook, NJ, USA). Stall swabs were collected using sponges soaked in neutralizing buffer (3M, St. Paul, MN, USA). Collection of environmental samples included the swabbing of the inside of the stall door, stall walls, rim of the feeder, and/or water bucket if available. Only two of the investigators (N.P. and C.J.F.) collected the stall swabs in order to collect samples in a consistent and systematic fashion. In order to prevent possible cross-contamination during sample collection and handling, the collection of nasal and stall swabs followed stringent biosecurity protocols. All investigators participating in the sample collection wore disposable coveralls, boots, and gloves. All samples were frozen at −80 °C for regulatory reasons and processed at the time the quarantine was lifted. 

All nasal swabs in viral transport medium and stall sponges in neutralizing buffer were processed for total nucleic acid purification using an automated nucleic acid extraction system (QIAcubeHT, Germantown, MD, USA) according to the manufacturer’s recommendations. Purified nucleic acids were assayed for the presence of EHV-1 genomic DNA (*gB* gene assay and D/N/H_752_ allelic discrimination assays) using a combined thermocycler/fluorometer (QuantStudio 5, Applied Biosystems, Foster City, CA, USA) as previously reported [4,12]. qPCR-positive EHV-1 results were reported qualitatively (positive or negative) and quantitatively. Positive and negative qPCR results were defined as the presence or absence of EHV-1 target genes, respectively, following 40 amplification cycles. Quantitative qPCR results for EHV-1 in nasal secretions were normalized against a housekeeping gene (*equine glyceraldehyde-3-phosphate dehydrogenase* gene) and expressed as number of *gB* target genes per million cells as previously reported [4]. Quantitative qPCR results for EHV-1 in stall swabs were expressed as number of *gB* target genes per µL of purified DNA as previously reported [27].

### 4.3. Data Analysis

Demographic information from the study horses was evaluated using descriptive analyses (mean, standard deviation, and median). All statistical analyses were performed using Stata Statistical Software (College Station, TX, USA).

## 5. Conclusions

In conclusion, the study results showed that a relatively small number of healthy horses were subclinically infected with EHV-1. While the level of EHV-1 shedding was variable, some of the subclinically infected horses shed EHV-1 at levels similar to clinically infected horses. Further, environmental contamination with EHV-1, measured via stall swabs, was an indirect indicator of temporal EHV-1 shedding of silently infected sport horses. The study results highlight the risk of subclinical EHV-1 shedders and their contaminated stalls as sources of infection for susceptible horses. The testing of individual stalls for the presence of EHV-1 may be a more practical approach than the collection of individual nasal swabs for the monitoring and early detection of the circulating virus. The results also highlight the need to improve the cleanliness and disinfection of stalls utilized by performance horses during show events. 

## Figures and Tables

**Figure 1 pathogens-11-00720-f001:**
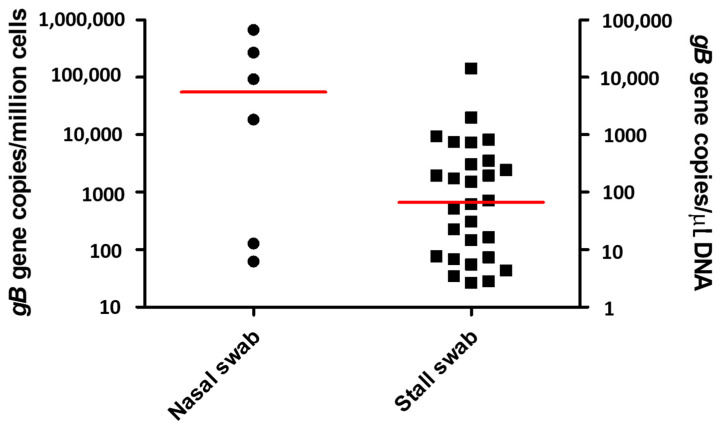
Absolute quantitation of EHV-1 expressed as number of target *gB* genes per million cells in nasal swabs from healthy shedders and as number of target *gB* genes per µL of purified genomic DNA for stall swabs. The horizontal red lines represent median values.

**Table 1 pathogens-11-00720-t001:** Qualitative and quantitative qPCR results for EHV-1 in nasal and stall swabs. The quantitative qPCR results are expressed as number of *gB* genes per million cells for nasal swabs and as number of target *gB* genes per µL of purified genomic DNA for stall swabs. Collection of environmental samples included the swabbing of the inside of the stall door, stall walls, rim of the feeder, and/or water bucket if available.

Barn Number	Horse/Stall ID	qPCR Results	
		Horse Nasal Swab (*gB* Genes/Million Cells)	Stall Swab (*gB* Genes/µL of Purified DNA)
1	305	Positive (92,751)	Negative
1	167	Negative	Positive (5.5)
1	169	Negative	Positive (51.27)
1	172	Negative	Positive (820.6)
1	174	Negative	Positive (14.6)
1	175	Negative	Positive (71.5)
1	178	Negative	Positive (4.3)
1	182	Negative	Positive (1979.2)
1	184	Negative	Positive (936.1)
1	185	Negative	Positive (2.8)
1	186	Negative	Positive (30.5)
1	188	Negative	Positive (306.6)
1	194	Horse moved to isolation	Positive (195.4)
2	24	Negative	Positive (14,274.9)
2	25	Negative	Positive (352.2)
2	27	Negative	Positive (7.7)
2	28	Negative	Positive (61.8)
2	30	Negative	Positive (729.4)
2	32	Negative	Positive (243.9)
2	34	Negative	Positive (195.4)
4	140	Negative	Positive (744.7)
4	148	Horse moved to isolation	Positive (6.8)
7	133	Positive (670,939)	Negative
8	200	Negative	Positive (2.6)
8	201	Negative	Positive (151.2)
32	153	Positive (62)	Negative
37	226	Negative	Positive (173.7)
39	314	Positive (268,960)	Negative
39	324	Positive (128)	Negative
45	105	Positive (18,442)	Negative
45	104	Negative	Positive (3.5)
45	106	Negative	Positive (22.5)
FEI II	240	Negative	Positive (16.5)
FEI III	257	Negative	Positive (7.3)

## Data Availability

Data are available on request due to privacy restrictions.

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
