# Peer review of "Molecular Monitoring of EHV-1 in Silently Infected Performance Horses through Nasal and Environmental Sample Testing"

_pathogens, 2022, doi:10.3390/pathogens11070720_

Round 1
Reviewer 1 Report
The article describes a study involving the silent shedding and environmental contamination of equine herpesvirus type 1 in the face of an outbreak. The apparent aim of the paper is to determine the frequency of silent EHV-1 shedders and their impact on environmental contamination. Both nasal swabs and stall swabs were collected from healthy horses that were attending an equestrian event that underwent quarantine due to an outbreak of EHV-1. The findings elucidated that silent infections of EHV-1 did occur with variable levels of shedding and that environmental contamination indirectly correlated with temporal silent EHV-1 shedding.
This manuscript will add great value to the veterinary community. It could be improved with a few clarifications.
I did not see an explicit definition of "silent transmission", "silent shedding", "subclinically infected horses" and the author's determination of lytic versus latent infections versus negative samples; clarifying these topics will help the readers understand the methods as well as the interpretation of the results.
Although the importance and aim of the study are clearly stated in lines 101-107, they were not overtly clear to me within the introduction. I did not see that sample handling was discussed in regards to mitigating contamination during sampling and during handling. The statistical tests used in the data analysis were not elucidated in the methods, which would enhance the interpretation of the results.
Author Response
This manuscript will add great value to the veterinary community. It could be improved with a few clarifications.
I did not see an explicit definition of "silent transmission", "silent shedding", "subclinically infected horses" and the author's determination of lytic versus latent infections versus negative samples; clarifying these topics will help the readers understand the methods as well as the interpretation of the results.
The authors thank the reviewer for pointing out important definitions regarding clinical and EHV-1 shedding status of the study horses. Silent was use as a surrogate for subclinical and better define an infected equid shedding EHV-1 in the absence of clinical sign. The term lytic was replace across the manuscript by replicating or contagious as lytic refers to in-vitro cell culture characteristics. Negative EHV-1 sample by qPCR refers to absence of detection of EHV-1 target gene by qPCR following 40 amplification cycles. The various definitions have been added under materials and methods.
Although the importance and aim of the study are clearly stated in lines 101-107, they were not overtly clear to me within the introduction. I did not see that sample handling was discussed in regards to mitigating contamination during sampling and during handling. The statistical tests used in the data analysis were not elucidated in the methods, which would enhance the interpretation of the results.
The aim of the study was added in the introduction. Information pertaining to good practices aimed at reducing the risk of contamination during sampling and during handling was added under materials and methods. Because of the small number of EHV-1 qPCR-positive samples, no statistical tests were performed. .
Reviewer 2 Report
The purpose of this study was to to investigate nasal and environmental stall swabs from a large group of healthy hunter and jumper show horses during a large EHM outbreak.
The study results highlight the risk of silent EHV-1 shedders and stalls occupied by subclinically infected horses as sources of infection for susceptible horses. The testing of individual stalls for the presence of EHV-1 may be a more practical approach than the collection of individual nasal swabs for the monitoring and early detection of circulating virus. The results also highlight the need to improve cleanliness and disinfection of stalls utilized by performance horses during show events.
Lines 44-45 : Please remplace « The highly contagious nature of EHM cases » by « The highly contagious nature of EHV-1 cases » because EHM is the pathology and not the pathogen. The pathology was not contagious.
Lines 46 : Please remplace « EHM» by « EHV-1» because EHM is the pathology and was not contagious .
In terms of methodology, the authors do not describe the statistical tests performed, only the name of the software.
I disagree with the term qPCR used throughout the article for the stall swab. Indeed, the abbreviation qPCR describes a quantitative PCR, which requires a quantification of the viral load as described in the previous articles cited by the authors with standard curve. In this manuscript, the authors present only Ct (Cycle threshold) values for the stall swabs, which is not quantitative. The authors could present their data in viral loads/µl of DNA extract or viral loads/ml of recovery solution of each swab. If it is not possible, remplace all « qPCR » by PCR for the data of stall swabs (lines 24,24,74,75,76,77,79,81,135, 136).
Line 200 : please remplace « QantSudio 5 » by « QuantSudio 5 »
In figure 1, authors indicate « The horizontal dashed line represents the laboratory specific threshold above which mRNA transcripts become detectable and EHV-1 lytic infection is suspected. ». What are the transcript expression levels in this setting? Has lytic infection been shown by cell culture for these 4 samples?
If as indicated lines 127-129, « 4/6 EHV-1 qPCR-127 positive nasal secretions would have contained lytic virus, representing a source of transmission to susceptible horses. », I don't understand how a highly excretory horse (horse 305, 133 and 314) can have negative environmental samples ? How can virus not be found in his immediate environment? What is the Ct value for these nasal swabs for the 6 positives horses ? Currently we cannot compare nasal swabs (viral load/million cells) and environmental samples (raw data, Ct value). Has lytic infection been shown by cell culture for the positive stall swabs with low Ct value (Ct<31)?
Further develop the discussion. There is no discussion between stalls with Ct<30 and infected horses that were isolated. What is the distance between the positive stalls and the stalls of the horses that were positive? Between the positive stalls and the positive horse without symptoms in the same barn? how can we overcome this?
What role could the staff have played in the contamination of other stalls in the same barn?
In table 1, please indicate the location of environmental samples (inside of the stall door, stall walls, rim of the feeder and/or water bucket)
Author Response
Lines 44-45 : Please replace « The highly contagious nature of EHM cases » by « The highly contagious nature of EHV-1 cases » because EHM is the pathology and not the pathogen. The pathology was not contagious.
As suggested by the reviewer, the change was made in the manuscript.
Lines 46 : Please replace « EHM» by « EHV-1» because EHM is the pathology and was not contagious .
As suggested by the reviewer, the change was made in the manuscript.
In terms of methodology, the authors do not describe the statistical tests performed, only the name of the software.
Because of the small number of EHV-1 qPCR-positive samples, no statistical tests were performed.
I disagree with the term qPCR used throughout the article for the stall swabs. Indeed, the abbreviation qPCR describes a quantitative PCR, which requires a quantification of the viral load as described in the previous articles cited by the authors with standard curve. In this manuscript, the authors present only Ct (Cycle threshold) values for the stall swabs, which is not quantitative. The authors could present their data in viral loads/µl of DNA extract or viral loads/ml of recovery solution of each swab. If it is not possible, replace all « qPCR » by PCR for the data of stall swabs (lines 24,24,74,75,76,77,79,81,135, 136).
As suggested by the reviewer, the data from the stall swabs has been converted in absolute values and reported as number of target genes (gB gene) per µl of purified DNA.
Line 200 : please replace « QantSudio 5 » by « QuantSudio 5 »
As suggested by the reviewer, the change was made in the manuscript.
In figure 1, authors indicate « The horizontal dashed line represents the laboratory specific threshold above which mRNA transcripts become detectable and EHV-1 lytic infection is suspected. ». What are the transcript expression levels in this setting? Has lytic infection been shown by cell culture for these 4 samples?
In previous work, the authors have shown that transcripts for the gB gene become detectable at a threshold of ≥ 1 x 104 gB genes/million cells. Transcripts represent molecular proxy for viable virus. In order to avoid any confusion and mostly because the samples were not cultured, the word “lytic” was changed to viable virus.
If as indicated lines 127-129, « 4/6 EHV-1 qPCR-127 positive nasal secretions would have contained lytic virus, representing a source of transmission to susceptible horses. », I don't understand how a highly excretory horse (horse 305, 133 and 314) can have negative environmental samples ? How can virus not be found in his immediate environment? What is the Ct value for these nasal swabs for the 6 positives horses ? Currently we cannot compare nasal swabs (viral load/million cells) and environmental samples (raw data, Ct value). Has lytic infection been shown by cell culture for the positive stall swabs with low Ct value (Ct<31)?
The authors speculate that a high viral load detected in nasal secretions of a subclinically infected horse reflect peracute infection. While these horses may have reached the peak of viral shedding in nasal secretions, there may not have been enough time elapsed to allow for EHV-1 buildup and detection via qPCR in environmental samples. The results for the stall swabs were, as suggested by the reviewer, changed into number of gB target genes per µl of purified DNA. Because lytic infection was not documented via cell culture, this limitation was added in the manuscript under discussion.
Further develop the discussion. There is no discussion between stalls with Ct<30 and infected horses that were isolated. What is the distance between the positive stalls and the stalls of the horses that were positive? Between the positive stalls and the positive horse without symptoms in the same barn? how can we overcome this?
What role could the staff have played in the contamination of other stalls in the same barn?
The reviewer brings up a very important point, which is the clustering of EHV-1 qPCR-positive stalls in the same barn and the point source for spread. The origin of EHV-1 is speculative but likely originated form a focal source and slowly spread thereafter directly via horse-to-horse contact or indirectly via contaminated equipment. The distance between EHV-1 qPCR-positive stall within the same barn ranged from as little as a few inches (adjacent stalls) up to 30 yards. The reviewer brings a very important point, which is contamination of other stall in the same barn by personal (rider, groom, stall cleaners/feeders, etc). The latter point was added in the discussion.
In table 1, please indicate the location of environmental samples (inside of the stall door, stall walls, rim of the feeder and/or water bucket)
As suggested by the reviewer, the change was made in the manuscript.
Reviewer 3 Report
The authors investigated the frequency of EHV-1 detection in healthy horses quarantined during a large equine herpesvirus myeloencephalopathy (EHM) outbreak. EHM has caused economic damage to the horse industry. The results reported in the present manuscript has a scientific significance for control EHM.
The authors detected EHV-1 nucleic acids using the protocol reported previously. In the previous reports, the authors' group clearly indicated gDNA and cDNA as the viral genomic DNA (gDNA) and complementary DNA (cDNA)prepared by reverse transcription of viral mRNA of gB gene, respectively. However, the authors did not indicate which type of nucleic acids were used for EHV-1 detection in nasal swabs and environmental swabs in the present manuscript. If the authors prepared cDNA of gB m RNA from the environmental sawbs, it is doubtful on the results reported in the present study. cDNA of gB has been used as a marker for lytic replication of EHV-1 in cells. It should be unable to detect gB mRNA in the environmental samples. Therefore the authors should improve descriptions concerning qPCR to indicate which type of DNA, gDNA or cDNA, was used as template DNA in lines 66, 74 to 83, 144, and 195 to 209.
Furthermore, the authors described in the explanation of Figure 1 that the horizontal dashed line represents the laboratory specific threshold above which mRNA transcripts become detectable and EHV-1 lytic infection is suspected. This description is unable to understand. Did EHV-1 replicated in the stalls and caused lytic infection without living cells on stalls?
A minor point. Line 106: "If" should be "It".
Author Response
The authors detected EHV-1 nucleic acids using the protocol reported previously. In the previous reports, the authors' group clearly indicated gDNA and cDNA as the viral genomic DNA (gDNA) and complementary DNA (cDNA) prepared by reverse transcription of viral mRNA of gB gene, respectively. However, the authors did not indicate which type of nucleic acids were used for EHV-1 detection in nasal swabs and environmental swabs in the present manuscript. If the authors prepared cDNA of gB m RNA from the environmental sawbs, it is doubtful on the results reported in the present study. cDNA of gB has been used as a marker for lytic replication of EHV-1 in cells. It should be unable to detect gB mRNA in the environmental samples. Therefore the authors should improve descriptions concerning qPCR to indicate which type of DNA, gDNA or cDNA, was used as template DNA in lines 66, 74 to 83, 144, and 195 to 209.
The authors apologize for any confusion that may have occurred in interpreting molecular results. All the analysis were performed using genomic DNA. Further, the interpretation of viable virus based on absolute quantitation at the level of genomic DNA is based on a previous study by the authors but was not performed in this study. The previously established threshold at the genomic DNA level used to establish replication was removed from Figure 1 in order to prevent any confusion. The authors have made the changes in the manuscript and highlighted that all analysis were performed at the genomic DNA level.
Furthermore, the authors described in the explanation of Figure 1 that the horizontal dashed line represents the laboratory specific threshold above which mRNA transcripts become detectable and EHV-1 lytic infection is suspected. This description is unable to understand. Did EHV-1 replicated in the stalls and caused lytic infection without living cells on stalls?
The authors realize that the statement may be confusing. Previous work by the authors has established molecular surrogate for replication in two ways: (i) presence of mRNA for the gB gene of EHV-1 and (ii) absolute quantitation with a viral load ≥ 1 x 104 gB genes/million cells at the genomic DNA level. To prevent any confusion, the authors have added that the previously established threshold used genomic DNA.
A minor point. Line 106: "If" should be "It"
As suggested by the reviewer, the change was made in the manuscript.
Reviewer 4 Report
This manuscript describes the molecular monitoring of EHV-1 through nasal and environmental sample testing. This measurement could potentially be very interesting to determinate the silent shedders during outbreaks and quarantines. It is really surprising the results showed in the table 1. All the horses positive to nasal swabs were negative to the stall swabs and all the positives for stall swabs were negatives for nasal swabs. Do the author think that could be only for the accumulative effects? Do you have any hypothesis why horses were positive just at one? I think the article is very interested and maybe can be clarify this point in the discussion.
Author Response
This manuscript describes the molecular monitoring of EHV-1 through nasal and environmental sample testing. This measurement could potentially be very interesting to determinate the silent shedders during outbreaks and quarantines. It is really surprising the results showed in the table 1. All the horses positive to nasal swabs were negative to the stall swabs and all the positives for stall swabs were negatives for nasal swabs. Do the author think that could be only for the accumulative effects? Do you have any hypothesis why horses were positive just at one? I think the article is very interested and maybe can be clarify this point in the discussion.
The authors speculate that a high viral load detected in nasal secretions of a subclinically infected horse reflect peracute infection. While these horses may have reached the peak of viral shedding in nasal secretions, there may not have been enough time elapsed to allow for EHV-1 buildup and detection via qPCR in environmental samples.
Round 2
Reviewer 1 Report
Thank you for your edits to this manuscript. This study provides interesting insight into another avenue for managing EHV-1 outbreaks.
Reviewer 2 Report
Authors made significant effort to address my comments.